# Delayed Topographical and Refractive Changes Following Corneal Cross-Linking for Keratoconus

**DOI:** 10.3390/jcm11071950

**Published:** 2022-03-31

**Authors:** Maryam Eslami, Farhad Ghaseminejad, Paul J. Dubord, Sonia N. Yeung, Alfonso Iovieno

**Affiliations:** Department of Ophthalmology and Visual Sciences, University of British Columbia, Vancouver, BC V5Z0A6, Canada; maryam.eslami@alumni.ubc.ca (M.E.); farhadg@alumni.ubc.ca (F.G.); paul@pjdubord.com (P.J.D.); sonia.y@gmail.com (S.N.Y.)

**Keywords:** corneal crosslinking, keratoconus, corneal aberrations

## Abstract

Background: The aim of this study was to analyze the long-term topographic and refractive outcomes of corneal cross-linking (CXL) in keratoconus. Methods: We used a retrospective observational study of patients with keratoconus who underwent CXL with a minimum follow-up of 5 years. Patients’ refractive and topography data (corrected distance visual acuity, sphere, cylinder, average and maximum keratometry, and corneal aberrations) were collected. Results: A total of 112 patients/150 eyes (mean age: 33.2 ± 10.7 years; range: 13–61) were included. The mean follow-up was 5.87 ± 1.35 years (range: 5–10). At the last follow-up visit, an improvement in CDVA, spherical and cylindrical refraction, average and steepest keratometry, and corneal aberrations were observed (*p* < 0.05), with the exception of trefoil. At the last visit, 49 (34.8%) and 31 (22.0%) eyes had an improvement beyond 1D in their spherical and cylindrical power, respectively, and 43 (28.7%) eyes had a flattening of their steepest keratometry. Progressive improvement over time was observed for spherical refraction; max and mean-K; as well as corneal RMS, total, high, coma, and spherical aberrations (*p* < 0.05). More severe disease at the baseline correlated with an improvement in corneal aberrations over time. Conclusions: In addition to a progressive improvement in refractive and keratometric indices, corneal aberrations also demonstrate a steady decline with long-term follow-up after CXL, which was more pronounced in more severe patients.

## 1. Introduction

Keratoconus is the most common primary corneal ectasia, affecting both genders and all ethnicities [1,2]. It may result in progressive and severe loss of vision [1,3], with younger patients demonstrating the greater progression of their disease [2] Management options include spectacles, contact lenses, intracorneal ring segments, Corneal Collagen Crosslinking, and ultimately lamellar or penetrating keratoplasty [1].

Corneal Collagen Crosslinking (CXL) is a procedure consisting of the controlled ultraviolet irradiation of the cornea, which has been pre-treated with a photosensitizer, riboflavin [3,4]. Deeper structures such as the lens and the retina are shielded from ultraviolet light by the stromal penetration of riboflavin [2]. This technique has been shown to increase corneal stiffness and is the only treatment modality that aims to stop the progression of keratoconus [4,5].

Since its first human clinical study in 2003 [5], various studies have looked at the long-term effect of CXL on refractive and topographic parameters [3,4,5,6,7,8,9]. These studies report variable amounts of stabilization or the improvement of distance-corrected visual acuity and keratometry values. However, all are limited by their small sample size, their length of follow-up, or both, with the vast majority of them being conducted in Europe. This study focuses on the long-term effects of CXL on corneal aberrations in addition to refractive and keratometric values, providing the largest sample size of patients with keratoconus to date, with up to 10 years of follow-up post-CXL in North America.

## 2. Materials and Methods

This is a retrospective review of patients with keratoconus who underwent CXL from January 2009 to January 2016 by two surgeons (SNY, PJD) at a single location. The study conformed to the provisions of the Declaration of Helsinki and was approved by the Institutional Research Ethics Board of the University of British Columbia.

### 2.1. Study Population

A prospective list was kept of all KC patients undergoing CXL from 2009 to 2016. All patients were considered and assessed for eligibility. The inclusion criteria were a clinical and topographical diagnosis of progressive KC, central corneal thickness of minimum 400 µm, and available refractive and topographic data for at least 5 years post-CXL. Progressive keratoconus was defined as increase of 1 D or more in the maximum keratometry value over 12 months. Patients were excluded if they had any adjunctive corneal interventions such as keratoplasty, intracorneal ring segments, or ocular comorbidities that may have impacted their refractive or topographic parameters.

### 2.2. Operative Technique

CXL was performed according to Wollensak et al.’s standard Dresden protocol [4]. After the instillation of topical anesthesia (0.5% proparacaine hydrochloride), corneal epithelium was manually debrided from the central 9 mm followed by the application of 0.1% riboflavin solution (10 mg riboflavin-5-phosphate in 10 mL dextran-T-500 20% solution) onto the cornea every 5 min for 30 min. The cornea was then exposed to UV light with a wavelength of 370 nm and an irradiance of 3 mW/cm^2^ for a total time of 30 min using the UV-X System (Peschke Meditrade CCL-365, Hunenberg, Switzerland) corresponding to a corneal exposure of 5.4 J/cm^2^.

### 2.3. Data Collection

Corrected distance visual acuity (CDVA, LogMAR), spherical and cylindrical refraction (diopters, D), mean keratometry (K-mean), maximum keratometry (K-max), corneal aberrations (total, high, coma, trefoil, and spherical), and root mean square (RMS) were collected prior to CXL, at 3 years post-CXL, at 5 years post-CXL, at 7 years post-CXL, and at the final available follow-up appointment.

Autorefraction, topography readings, and corneal aberrations were obtained from the NIDEK OPD Scan III (Nidek Technologies, Gamagori, Japan). Corneal wavefront aberrations were reconstructed using a sixth-order Zernike polynomial decomposition for a 4 mm pupil size. Due to a change in equipment, no corneal wavefront aberrations were available prior to May 2012.

Changes in sphere, cylinder power, and keratometry were calculated at each time point, as well as the last available data from the pre-CXL value. Topographic stability was defined as a change in keratometry of less than 1 D, while steepening and flattening were defined as an increase or decrease over 1 D, respectively. The change in CDVA presented in LogMAR was also calculated between the last available data and the pre-CXL value, for which 0.1 was used as the cut-off for stability, improvement, and worsening.

### 2.4. Data Analysis

Data were expressed as mean ± standard deviation. A paired Student’s *t*-test was used to compare the last available data to the pre-CXL value. Due to the retrospective nature of the study with a declining sample size past the 5-year follow-up, a comparison analysis was performed between the available and missing datapoints to ensure statistical equivalence. A comparison of the measurements at different time points was performed using a repeated measures analysis of variance. A correlation analysis between the data at presentation and the change in corneal aberrations was also carried out. *p* value less than 0.05 was considered statistically significant.

## 3. Results

One hundred and twelve patients (150 eyes) met the eligibility criteria and were included in the study. The mean age was 33.2 ± 10.7 years (median: 34; range: 13–61) and 78 patients (69.6%) were male. The mean follow-up time was 5.87 ± 1.35 years (median: 5.75; range 5–10). Table 1 summarizes the refractive and topographic study measures over time. The *t*-test subgroup comparison of the missing and available datapoints at the 3 yrs, 5 yrs, and 7 yrs follow-ups showed no significant difference in the baseline data.

### 3.1. Refractive Results

CDVA improved from 0.268 ± 0.172 LogMar pre-CXL to 0.228 ± 0.160 at the last follow-up visit (*p* = 0.020).

Spherical power also improved from −6.72 ± 4.12 D pre-CXL to −6.17 ± 4.26 D at the last follow-up visit (*p* < 0.001). Similarly, cylinder power decreased from 3.42 ± 2.23 D to 2.98 ± 1.84 D (*p* < 0.001). The sphere and cylinder power trend over time is depicted in Figure 1. A repeated measure analysis of variance showed a statistical significance for the improvement of sphere power over time (*p* < 0.05). A similar trend was not observed for cylinder (Figure 1A). The spherical equivalent at presentation was −4.79 ± 3.61 D, which declined to −4.57 ± 3.75 D at the last follow-up visit. This trend was not statistically significant (*p* = 0.30). At the last follow-up visit, 71 (50.4%) eyes had stable spherical power, 49 (34.8%) eyes had improved beyond 1 D, and 21 (14.9%) eyes had worsened more than 1 D. However, the cylindrical power showed a higher stability rate, with 94 (66.7%) eyes remaining within 1D of their pre-CXL value and 31 (22.0%) eyes decreasing beyond 1 D (Figure 1B). The highest observed regression of myopia and astigmatism was 8.75 and 5 D, respectively.

### 3.2. Topographic Results

The average mean-K value was 46.68 ± 2.92 D preoperatively and 46.10 ± 3.04 D at the last follow-up visit. Similarly, the average max-K values were 48.35 ± 3.40 D and 47.64 ± 3.41 D, respectively. The average change in mean-K and max-K at the last follow-up visit were −0.58 ± 1.30 D (range −5.79 to 2.39 D) and −0.71 ± 1.59 D (range −7.27 to 3.36 D), respectively (*p* < 0.0001). A repeated measure analysis of variance showed a progressive improvement in both mean-K and max-K over time as well. Figure 2A illustrates the change of the keratometry values from their pre-CXL baseline at each time point.

At the last visit, 99 (66%) eyes showed a stability of mean-K, 43 (28.7%) eyes had flattening, 8 (5.3%) eyes had steepened beyond 1 D, while 20 (13.3%) eyes had flattened, and 3 (2%) eyes had steepened beyond 2 D. A total of 95 (63.3%) eyes showed a stability of max-K, 48 (32.0%) eyes had flattening, and 7 (4.7%) eyes had steepening. A total of 28 (18.7%) eyes had flattening and only 3 (2%) eyes had a steepening of max-K beyond 2 D. The keratometric stability at each time point is illustrated in Figure 2B, with an increasing percentage of eyes showing a flattening of both Max-K and Mean-K over time. There was no correlation between the changes in topography indices over time and the observed amelioration of CDVA.

### 3.3. Corneal Aberrations

Corneal aberration data were unavailable prior to May 2012; therefore, 82 eyes were included in this analysis. The change in corneal aberration and the RMS of eyes with available pre-CXL data are shown in Figure 3. There was a statistically significant reduction in all corneal wavefront aberrations and corneal RMS at the last follow-up visit compared to the preoperative values, with the exception of trefoil. The total corneal aberration was reduced from 7.85 ± 6.21 pre-CXL to 5.92 ± 4.58 (*p* < 0.001) at the last follow-up visit. Similarly, higher-order aberration decreased from 2.38 ± 1.74 to 1.88 ± 1.40 (*p* < 0.001), coma from 2.04 ± 1.65 to 1.52 ± 1.33 (*p* < 0.001), trefoil from 0.81 ± 0.56 to 0.72 ± 0.51 (*p* = 0.057), spherical from 0.45 ± 0.47 to 0.33 ± 0.35 (*p* = 0.003), and corneal RMS from 2.39 ± 1.74 to 1.89 ± 1.40 (*p* < 0.001). A similar trend was observed over time with a repeated measure analysis of variance (Figure 3).

A statistically significant correlation was found between the improvement in corneal aberrations (total, high, coma, sphere, and corneal RMS) and baseline spherical refraction, higher max-K, mean-K, corneal RMS, total, high, coma, and spherical aberrations.

There were 16 eyes with available corneal aberration and visual acuity data pre-CXL and 7 years post-CXL. In this small cohort of patients, a correlation between improvement in CDVA and improvement in corneal RMS, coma, and high-order aberrations was observed.

## 4. Discussion

There are numerous studies in the literature reporting the short-term outcomes (12–24 months) of CXL in keratoconus patients. However, there are only a few studies with a follow-up longer than 4 years [3,4,6,7,8,10,11], and with a limited sample size of a maximum of 44 eyes of adult patients [4] and 62 eyes of pediatric patients [11]. This study provides long-term refractive and topographic data post-CXL on 150 eyes with a minimum of 5 years and up to 10 years of follow-up, which is the longest follow-up of North American CXL patients reported. The results, while in accordance with the main points of previous reports regarding the stabilization of keratoconus, provide useful insight into the long-term trajectory of CXL outcomes. After a minimum follow-up of 5 years post-CXL, our series shows a progressive flattening of keratometry values over time and a regression of myopia and astigmatism, as well as a reduction in corneal aberrations.

The improvement in CDVA of 0.06 LogMAR observed at the 3-year follow-up in this study was statistically significant and remained the same at the 5-year point. A similar plateau has been observed in other studies with longer than 4 years of follow-up [8,11]. The 69 eyes with a 7-year follow-up demonstrated a non-statistically significant gain of 0.073 LogMAR, as the pre-CXL CDVA was 0.285 ± 0.173. Varying improvement in CDVA is reported in the literature [12]. Raiskup-Wolf et al. reported a gain of 0.14 LogMAR after 10 years of follow-up in 34 eyes; however, their pre-CXL mean CDVA was 0.4 LogMAR [10] (0.13 LogMAR higher than our study). Another study of 39 eyes with a 5-year follow-up showed a 0.11 LogMAR gain with a pre-CXL CDVA of 0.31 LogMAR [8]. Similarly, Wittig-Silva et al. reported a gain of 0.09 LogMAR in 48 eyes after 3 years with a baseline CDVA of 0.33 LogMAR [9]. The gain in CDVA observed in this study (0.04 LogMAR) was smaller than that seen in the mentioned reports; however, our eyes started with better CDVA pre-CXL, therefore limiting the degree of improvement possible.

There was a statistically significant improvement in both the sphere and cylinder power from the pre-CXL value to the last follow-up visit in our study. There are variable reports in the existing literature of improvements in spherical and cylindrical error of up to 1.87 D and 4.62 D, respectively [3,4,6,7,8,9,10,11]; however, most did not reach statistical significance likely due to the insufficient power of the studies. Our results also demonstrate that more than 85% of the eyes had stability or an improvement of their refractive sphere and cylinder power at their last visit.

Since the first report of CXL’s efficacy in halting the progression of keratoconus [5], many studies have looked at the topographic outcomes of patients post-CXL, reporting variable degrees of stability or improvement in keratometry values. Wollensak et al. [5], Raiskup-Wolf et al. [3], Caporossi et al. [4], and Kymionis et al. [7] all found changes in max-K of about 2 D. This is not in accordance with our study (average max-K change of 0.71 D at the last visit) (see Wittig-Silva et al. [9] (change of 1.03 D), or Hashemi et al. [8] (non-significant change of 0.24 D)). All the former studies had a higher pre-CXL max-K (54.2, 53.7, approximately 55, and 52.53 D, respectively), whereas the mean pre-CXL max-K in this study was 48.35 D. The higher regression of max-K in patients with more advanced keratoconus and higher baseline keratometry values has also been suggested by Wittig-Silva et al. [9]. However, the mentioned studies are all limited by their sample size and follow-up length. Yet, what is in accordance with prior studies is the high proportion of patients with stability or improvement in their disease. The failure rate (a steepening of max-K of 1 D or more) of 4.7% in adults in this paper is also similar to that of other published reports [10,13,14], while much lower than pediatric studies with a failure rate of up to 20% after 10 years of follow-up [11].

More recently, the observation of corneal flattening post-CXL has sparked interest in its potential use in the treatment of mild myopic refractive error [15]. The early data on this procedure, photorefractive intrastromal cross-linking (PiXL), have shown it to be effective and safe [15,16,17]. While the PiXL protocol and patient population are not identical to CXL, the long-term outcomes and the progressive corneal flattening and regression of refractive error occurring beyond 3 years of follow-up may also be applicable and of interest in relation to the PiXL procedure.

One of the least commonly reported outcomes following CXL is a change in wavefront aberrations after long-term follow-up. In this study, we found a statistically significant reduction in corneal RMS and total, high, coma, and spherical aberrations over time and at the last follow-up visit. Only 82 eyes were considered for an analysis of wavefront aberrations due to a lack of pre-CXL aberration data for procedures occurring prior to May 2012. Despite this limitation, this study still provides the largest cohort with a longer follow-up compared to previous studies. Vinciguerra et al. showed a reduction in coma and total aberrations after 12 months in 28 eyes [18], while Caporossi et al. demonstrated a reduction in higher-order aberrations and coma after 4 years in 44 eyes [4]. In this study, trefoil was the only aberration that did not demonstrate a statistically significant reduction. This finding was also reported by O’Brart et al.’s study of 30 eyes with 4–6 years of follow-up [6].

A more significant Improvement in corneal aberrations over time was detected in patients with higher baseline corneal aberrations, mean and max keratometry, and spherical refraction. In other words, patients with more severe keratoconus at baseline were more likely to experience an improvement in corneal aberrations. On the contrary, patients with more advanced keratoconus did not seem to experience more topographical corneal flattening during the follow-up time.

This prompted us to investigate in our series a possible correlation between an improvement in corneal aberrations and corrected visual acuity, as previously reported by El-Massry et al. [19]. If such an association was detected, one could have concluded that improved corneal aberrations, more than topographical flattening, would play a significant role in improving CDVA in the long term after CXL. In our study, such an association was only observed in a small cohort of patients 7 years after CXL. This association was not observed by Greenstein et al. [20], Naderan and Jahanrad [21], or Ghanem et al. [22], who used a shorter follow-up time. When we observed the trend of aberrations over time (Figure 3), a significant improvement was only displayed starting from about 5 years after CXL. As a consequence of that, a long period of time after CXL (>5 years) may be needed to observe the positive effect of the improvement in corneal aberrations on visual acuity.

This study’s main limitation is its retrospective design. There was also a reduced sample size past 5 years. Efforts were made to account for this limitation by a comparison analysis of the available and missing datapoints, as described. The change in equipment used during the study duration also reduced the number of eyes and the length of follow-up for corneal aberrations, as only CXL procedures performed after May 2012 could be considered for that analysis.

In summary, the long-term analysis of CXL outcomes in patients with keratoconus displayed an improvement in visual, topographic, and refractive outcomes. Corneal aberrations improve more significantly in advanced patients and may play a more significant role than corneal topography in determining visual improvement after CXL. These findings from a large cohort of patients provide new insight into the role of CXL in the management of KC, which should be taken into account when planning treatment strategies.

## Figures and Tables

**Figure 1 jcm-11-01950-f001:**
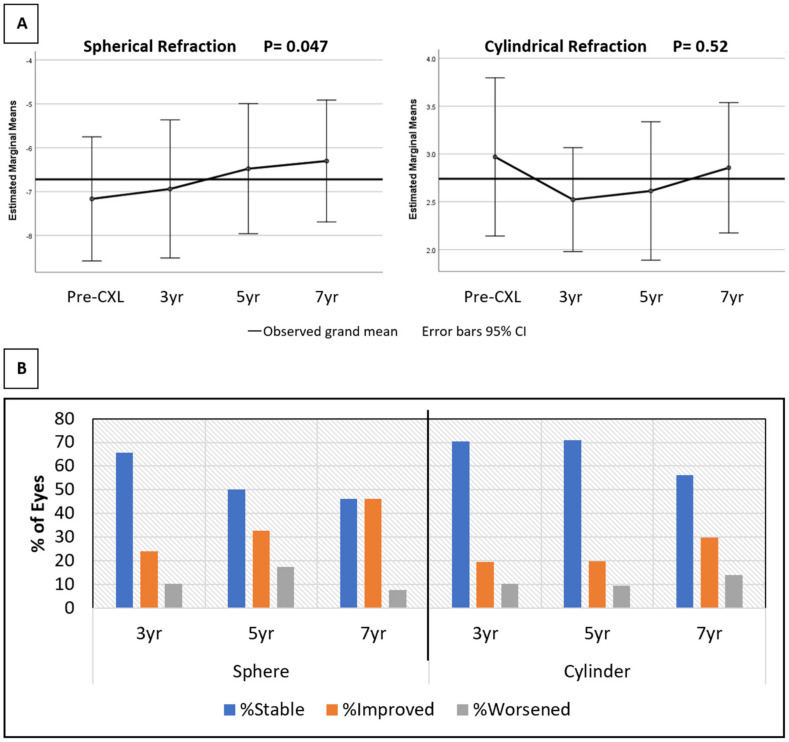
(**A**,**B**) Change in spherical and cylindrical power from the pre-crosslinking (CXL) value. (**B**): Percentage of eyes with stable (within 1 D), improved (<−1 D), and worsened (>1 D) refractive values over time.

**Figure 2 jcm-11-01950-f002:**
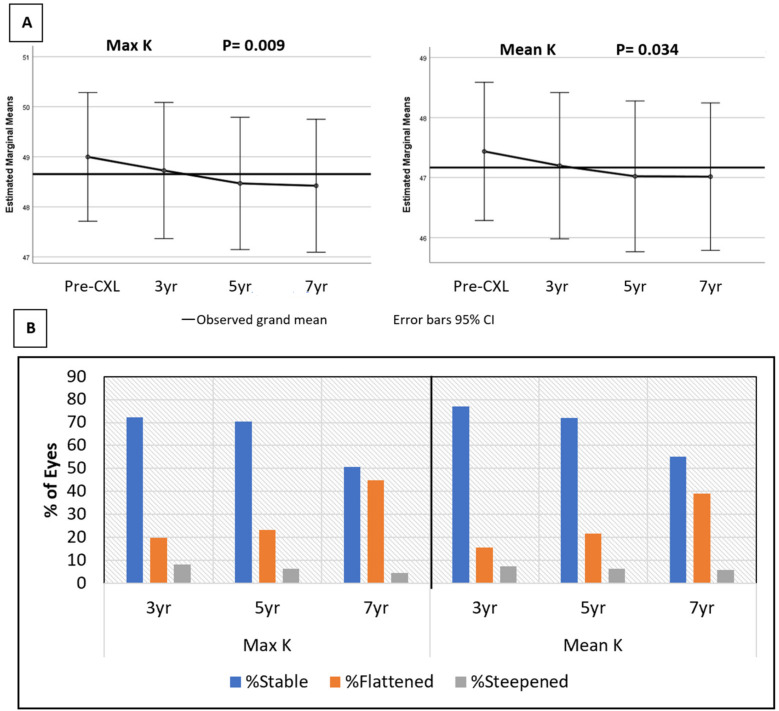
(**A**,**B**) Change in maximum and mean keratometry from the pre-crosslinking (CXL) value. (**B**): Percentage of eyes with stable (within 1 D), flattened (<−1 D), and steepened (>1 D) keratometry values over time.

**Figure 3 jcm-11-01950-f003:**
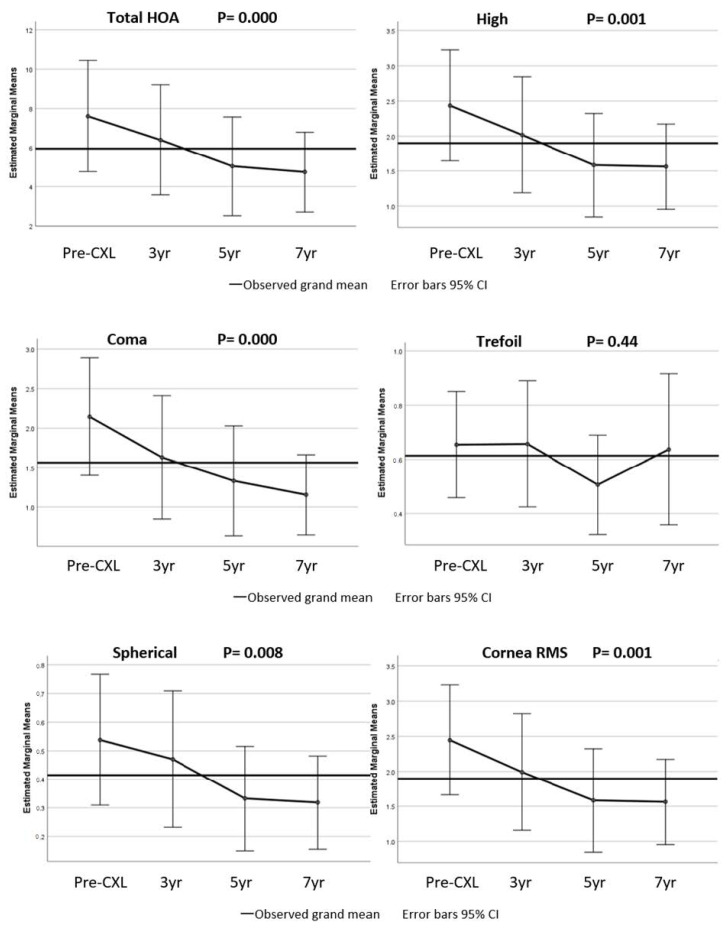
Change in corneal wavefront aberrations and corneal RMS from the pre-crosslinking (CXL) value.

**Table 1 jcm-11-01950-t001:** Study measures over time. Values are presented as mean ± standard deviation.

	**Pre-CXL** ** *n* ** **: 150**	**3 yrs** ** *n* ** **: 122**	**5 yrs** ** *n* ** **: 125**	**7 yrs** ** *n* ** **: 69**	**Last Available** ** *n* ** **: 150**	***p*** **Value ***
CDVA (LogMar)	0.268 ± 0.172	0.210 ± 0.139	0.215 ± 0.149	0.242 ± 0.145	0.228 ± 0.160	0.020
Sphere Power (D)	−6.72 ± 4.12	−6.22 ± 4.11	−6.28 ± 4.07	−5.80 ± 4.30	−6.17 ± 4.26	<0.001
Cylinder Power (D)	3.42 ± 2.23	3.04 ± 2.15	3.01 ± 1.95	2.99 ± 2.11	2.98 ± 1.84	<0.001
Mean-K (D)	46.68 ± 2.92	46.34 ± 2.93	46.36 ± 3.07	45.96 ± 3.38	46.10 ± 3.04	<0.001
Max-K (D)	48.35 ± 3.40	47.85 ± 3.43	47.89 ± 3.37	47.44 ± 3.68	47.64 ± 3.41	<0.001
	**Pre-CXL** ** *n* ** **: 82**	**3 yrs** ** *n* ** **: 108**	**5 yrs** ** *n* ** **: 112**	**7 yrs** ** *n* ** **: 69**	**Last Available** ** *n* ** **: 150**	***p*** **Value ***
Total Corneal Aberrations	7.85 ± 6.21	8.16 ± 7.07	7.06 ± 5.52	5.29 ± 4.17	5.92 ± 4.58	<0.001
High	2.38 ± 1.74	2.43 ± 1.98	2.15 ± 1.54	1.71 ± 1.25	1.88 ± 1.40	<0.001
Coma	2.04 ± 1.65	2.09 ± 1.89	1.79 ± 1.45	1.40 ± 1.16	1.52 ± 1.33	<0.001
Trefoil	0.81 ± 0.56	0.76 ± 0.57	0.76 ± 0.52	0.61 ± 0.43	0.72 ± 0.51	0.057
Spherical	0.45 ± 0.47	0.49 ± 0.58	0.42 ± 0.51	0.35 ± 0.41	0.33 ±0.35	0.003
Cornea RMS	2.39 ± 1.74	2.43 ± 1.98	2.15 ± 1.54	1.73 ± 1.24	1.88 ± 1.40	<0.001

(CDVA: corrected distance visual acuity; CXL: corneal cross-linking; D: diopters; Max-K: maximum keratometry; Mean-K: mean keratometry; LogMAR: logarithm of minimum angles of resolution; RMS: root mean square.) * Paired *t*-test between pre-CXL and the last available datapoint.

## Data Availability

Not applicable.

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
