# Peer review of "Delayed Topographical and Refractive Changes Following Corneal Cross-Linking for Keratoconus"

_jcm, 2022, doi:10.3390/jcm11071950_

Round 1

Reviewer 1 Report

For authors:

#1 The range of the sample presents a great dispersion between 13 and 61, the disease does not evolve the same in adolescents as in those older than 40 years.
#2 A total of 112 patients and 150 eyes, not including only one eye per patient could mask the results of the study in terms of evolution.
#3 It would be interesting to be able to know the values taking into account age and without bias. There is also no difference between men and women, I don't know if because there are no differences, I should notify you.

Reviewer 2 Report

Line 27. It reads: “Keratoconus is the most common primary corneal ectasia, affecting both genders and all ethnicities [1,2]”.

Comment: The cited reference # 1, is more than 10 years old. Recently Rabinowitz et al. and Santodomingo-Rubido et al. have published reviews on keratoconus.

Line 38. It reads: “Since its first human clinical study in 2003 [5], many studies have looked at the long-term effect of CXL on refractive and topographic parameters [3-9].”

Comment: In reality, relatively few studies have analyzed long-term results after crosslinking. Most of the studies have follow-ups of 3 years or less (in fact, the cited study by Wittig-Silva et al. included only 9 eyes with more than 1 year of follow-up, so it should be eliminated since it is not a long-term study).

Suggested additional references from long-term studies (at least 5 years minimum follow-up):

Galvis V, Tello A, Carreño NI, Ortiz AI, Barrera R, Rodriguez CJ, Ochoa ME. Corneal Cross-Linking (with a Partial Deepithelization) in Keratoconus with Five Years of Follow-Up. Ophthalmol Eye Dis. 2016 May 12;8:17-21. doi: 10.4137/OED.S38364. PMID: 27199574; PMCID: PMC4869599.

Nicula CA, Rednik AM, Nicula AP, Bulboaca AE, Nicula D, Horvath KU. Comparative Results Between "Epi-Off" Accelerated and "Epi-Off" Standard Corneal Collagen Crosslinking-UVA in Progressive Keratoconus - 7 Years of Follow-Up. Ther Clin Risk Manag. 2021 Sep 7;17:975-988. doi: 10.2147/TCRM.S321410. PMID: 34522101; PMCID: PMC8434931.

Line 70.

It reads: “Corrected distance visual acuity (CDVA, LogMAR), spherical and cylindrical refraction (diopters, D), mean keratometry (K-mean), maximum keratometry (K-max)…”

Comment:

Although the separate analysis of the spherical and cylindrical component of refraction is useful in certain analyses, the analysis of conjugate variables, which give a comprehensive idea of the refractive error of each eye, is undoubtedly necessary. The analysis must then also include at least the analysis of the spherical equivalent, and could also include that of the defocus equivalent.

Table 1 headers include the “N” letter to indicate the sample size.

Comment:

Although there is a discussion about it, it is generally accepted that the uppercase "N" refers to the number of sampling units making up the universe, and the lowercase "n" is used to refer to the number of sampling units in each sample itself. Therefore, in the header of each column, the "n" must be used in lowercase.
